# System-environmental entanglement in critical spin systems under $ZZ$-decoherence and its relation to strong and weak symmetries

Yoshihito Kuno[1*], Takahiro Orito[2] and Ikuo Ichinose[3†]

**1** Graduate School of Engineering science, Akita University, Akita 010-8502, Japan
**2** Institute for Solid State Physics, The University of Tokyo, Kashiwa, Chiba, 277-8581, Japan
**3** Department of Applied Physics, Nagoya Institute of Technology, Nagoya, 466-8555, Japan

* kuno421yk@gmail.com

## Abstract

Open quantum many-body systems exhibit nontrivial behavior under decoherence. In particular, system-environmental entanglement (SEE) is one of the efficient quantities for classifying mixed states subject to decoherence. In this work, we investigate the SEE of critical spin chains under nearest-neighbor $ZZ$-decoherence. We numerically show that the SEE exhibits a specific scaling law, in particular, its system-size-independent term ("$g$-function") changes drastically its behavior in the vicinity of phase transition caused by decoherence. For the XXZ model in its gapless regime, a transition diagnosed by strong Rényi-2 correlations occurs as the strength of the decoherence increases. We determine the location of the phase transition by investigating the $g$-function that exhibits a sharp change in the critical region of the transition. Furthermore, we find that the value of the SEE is twice that of the system under single-site $Z$-decoherence, which was recently studied by conformal field theory. From the viewpoint of Rényi-2 Shannon entropy, which is closely related to the SEE at the maximal decoherence, we clarify the origin of this $g$-function behavior.

## 1   Introduction

In practical systems, pure quantum states are exposed by environment, and emergent decoherence produces inevitable physical effects on the pure state. In most of studies searching novel quantum many-body states, it is assumed that the many-body system is isolated and is not affected by environment. In the research on quantum computers and quantum memories, the effect of decoherence by environment [1] is an important research subject. In quantum devices such as topological quantum memory [2–5] and noisy-intermediate-scale quantum computer [2,6,7], decoherences generate undesired effects on stored quantum information. On the other hand, the interplay between environment and quantum many-body system can lead to nontrivial quantum phase transitions and critical phenomena, which are not observed in isolated quantum systems. Recently, mixed states with no direct counterparts of pure states have attracted lots of attention. As an example, topologically-ordered states as well as symmetry protected topological pure states [8,9] change to nontrivial mixed states with another type of topological order by decoherence, the origin of which is the effect of environment [10–17]. Furthermore, recent studies are discussing and clarifying symmetries of the mixed state and their spontaneous symmetry breaking (SSB) emerging from decoherence. It has been clarified that there are various types of SSBs in the mixed state, such as strong symmetry SSB, weak symmetry SSB, strong-to-weak SSB (SWSSB) and strong-to-trivial SSB, etc, [11,12,18–28], and some of mixed states have a nontrivial long-range order (LRO). Discovering novel types of SSBs as well as a measure for observing them is an on-going important issue in quantum information and condensed matter communities.

(1-2)Furthermore, there is growing interest not only in various SSBs realized in mixed states, but also in the precursor phenomena emerging from the influence of the environment on critical states. We would like to understand how a mixed state survives as a critical state or alternatively how its universality class is changed by the environment. In particular, some of critical quantum states under decoherence are an interesting playground to investigate mixed state quantum phase transition and its criticality. As concrete examples, the recent works [29,30] studied mixed state phase transitions in critical spin chains by the conformal field theory (CFT), which are induced by local on-site decoherence. In Refs. [29,30], system-environmental entanglement (SEE) was studied and analytically examined to find that SEE exhibits an interesting scaling law including a universal term independent of the system size called "$g$-function" [31]. This universal term characterizes infra-red properties of mixed states.

(1-2)This approach, based on the observation of the SEE, highlights the existence of universality classes of critical mixed states that are distinct from those of pure states.

In this work, instead of on-site decoherence examined in [29,30], we shall study the effects of a multi-site decoherence of $ZZ$-type on critical states in the transverse-field Ising model (TFIM) and XXZ model. These states are described by $c = 1/2$ and $c = 1$ CFTs, respectively. (1-2)(1-1b)(2-1) The reason for choosing $ZZ$-type decoherence is that this nearest-neighbor decoherence can induce SWSSB. Moreover, study of that decoherence elucidates the possibility that the interplay between the order of the SWSSB and critical properties of the mixed states generates a new kind of universality class. In fact, the previous studies showed that this type of decoherence can induce nontrivial LROs and long-range entanglement for mixed states, e.g., SWSSB states [11,12,18–28].

In this setup, we study the following issues:

1. For the critical ground states of the TFIM and XXZ models, is $ZZ$-decoherence relevant? In particular, we are interested in how the decoherence transfers the initial critical state into a critical mixed state and if nontrivial behavior of SEE emerges there.

2. If mixed state in the above models changes non-trivially under $ZZ$-decoherence, does the SEE exhibit the scaling law proposed in the previous works, $S_{SE} = \alpha L - s_0 + \mathcal{O}(L^{-1})$ [29,30]? If this scaling law holds, how the universal term $s_0$ (characterizing low-energy property of the system) differs from that of the system under the on-site decoherence studied in [29,30]? If there is a difference, what is the origin of that?

3. Is there any relationship in behavior between the $g$-function and a symmetry order parameter of emerging mixed states? In particular, how a measure of entanglement and symmetry order parameters are related to each other?

To answer the above questions, we employ the doubled Hilbert space formalism [32,33] and filtering methods [34,35] in addition to numerical approach by using matrix product state (MPS). In this numerical approach, the SEE is efficiently calculated from the norm of the filtered MPS defined on ladder spin systems. By using the numerical methods, we find that $ZZ$-decoherence for the critical states in both the critical TFIM and XXZ models induces novel critical mixed phases with strong Rényi-2 correlation, and that the SEE exhibits the scaling law expected in [29,30,36]. However, interestingly enough, the universal term of the SEE (or $g$-function) takes different values from that of the single-site decoherence considered in [30,36].

For the critical TFIM case, we numerically observe that the SEE exhibits the expected scaling law, and the value of the $g$-function under the maximal decoherence is related to the value obtained from the Rényi-2 Shannon entropy [37,38].

As a more interesting result, we numerically show that for the critical XXZ model, the $g$-function for the strong $ZZ$-decohered mixed state is twice that of the system under a local on-site $Z$-decoherence [30]. We analytically explain the origin of this behavior of the $g$-function by considering the maximal decoherence limit and using the Rényi-2 Shannon entropy.

The rest of this paper is organized as follows. In Sec. 2, we introduce two critical spin-1/2 systems and $ZZ$-decoherence as a channel applied to the critical states of the models. In Sec. 3, the SEE is introduced, which is one of the target physical quantities in this work. In Sec. 4, we explain the doubled Hilbert space formalism to investigate the critical states subject to decoherence. There, the decoherence can be regarded as the filtering operation to the doubled critical states defined on the ladder spin system, which is explained somewhat in detail. In Sec. 5, we perform the systematic numerical calculations by using the MPS and the filtering method for both the TFIM and XXZ models. Some correlation functions [19,20, 39] to observe the strong or weak symmetry SSB are introduced and numerically calculated. Simultaneously, the SEE is investigated. In particular, the universal $g$-function denoted by $e^{s_0}$

is extensively studied. In Sec. 6, we discuss the physical meaning of the universal scaling law and the $g$-function $e^{s_0}$ in the SEE obtained numerically. Section 7 is devoted to the summary and conclusion.

## 2   Critical system and decoherence

In this work, we study effects of decoherence applied to critical states of two 1D spin systems, represented by Pauli operators $X_j$, $Y_j$ and $Z_j$. The first system is the 1D TFIM, Hamiltonian of which is given by

$$H_{\text{TFI}} = -\sum_{j=0}^{L-1}\left[ Z_j Z_{j+1} + X_j \right],$$

and the second one is the 1D XXZ model, which is given by

$$H_{\text{XXZ}} = \sum_{j=0}^{L-1}\left[ X_j X_{j+1} + Y_j Y_{j+1} + \Delta Z_j Z_{j+1} \right],$$

where $\Delta$ is anisotropic parameter. Throughout this work, periodic boundary conditions are imposed. Both models possess a $Z_2$ symmetry, which is nothing but the global spin flip $U_X = \prod_{j=0}^{L-1} X_j$. For the model $H_{\text{XXZ}}$, the critical ground state appears for $|\Delta| < 1$ described by Tomonaga-Luttinger Liquid (TLL) with the TLL parameter $K = \frac{\pi}{2(\pi-\arccos\Delta)}$, where $K > 1/2$. It is known that the critical ground states in the above models are described by $c = 1/2$ and $c = 1$ CFTs, respectively [40].

In this work, we study the effects of system-environment interactions, in particular, $ZZ$-decoherence applied to the critical state $\rho = |\phi_0\rangle\langle\phi_0|$, where $|\phi_0\rangle$ stands for the pure ground state at criticality in the TFIM or XXZ model. This kind of decoherence is induced by the interactions between the system and environment, such as $\rho_{\text{SE}} = \hat{U}(\rho \otimes \rho_{\text{E}})\hat{U}^\dagger$, where $\rho_{\text{E}}$ is density matrix of environment and $\hat{U}$ is the unitary operator representing the interactions between the system and environment. After tracing out the degrees of freedom of the environment, we obtain the system density matrix subject to the resultant decoherence.

Description of this decoherence by channel is given by [41]

$$\mathcal{E}_{tot}^{ZZ}[\rho] \;=\; \left(\prod_{j=0}^{L-1}\mathcal{E}_j^{ZZ}\right)[\rho] \equiv \rho_D, \tag{1}$$

$$\mathcal{E}_j^{ZZ}[\rho] \;\equiv\; (1-p_{zz})\rho + p_{zz}Z_j Z_{j+1}\rho Z_{j+1}Z_j, \tag{2}$$

where the strength of the decoherence is tuned by $p_{zz}$ ($j$-independent), and $0 \le p_{zz} \le 1/2$. (1-1a)(2-2) This channel can be expressed by the Lindbladian time-evolution dynamics. It is given by $\frac{d\rho}{dt} = \sum_j \frac{1}{2}[Z_j Z_{j+1}\rho Z_{j+1}Z_j - \rho]$ with a time interval $t = -\ln(1-2p)$. The local Lindbladian operation in the sum on the right-hand side can be implemented in experiments for quantum circuits, as proposed in [42, 43].

(1-1b)(2-1)Here, we explain the motivation for choosing the $Z_j Z_{j+1}$ operator as decoherence. This choice comes from the consideration on the symmetry of the system: it preserves the global $Z_2$ symmetry of the TFIM and XXZ models. It is known that the decoherence can induce or probe the SWSSB as recently discussed in the context of unconventional mixed state [11, 12, 18–28]. The simplest and most representative operation respecting the symmetry is the nearest-neighbor $ZZ$ operator. We are interested in how the global $Z_2$ symmetry is realized under the symmetry-preserving decoherence.

For $p_{zz} = 1/2$, the channel $\mathcal{E}^{ZZ}_{tot}$ corresponds to the projective measurement of $Z_j Z_{j+1}$ for a link between $j$- and $j + 1$-th sites without monitoring outcomes, which is called maximal decoherence. As we explained in the introduction, the reason why we consider $ZZ$-decoherence is that it can give an insight into how system-environment entanglement and strong/weak $U_X$ symmetry of the system [44] are related with each other [1]. Besides the SEE, we are also interested in how the above mentioned symmetries of non-trivial mixed states emerge as the strength of the decoherence is increased. To search a decoherence phase transition of the mixed state $\rho_D$ is also a target of the present study.

## 3   System-environmental entanglement

This study focuses on the SEE for a density matrix $\rho$ [29, 30] given by

$$S_{SE} = -\log \text{Tr}[\rho^2]. \tag{3}$$

This is also called the second-order Rényi entropy for the density matrix $\rho$. The SEE captures the degree of mixing of non-trivial mixed states induced by decoherences [29], and it can be diagnostic for mixed-state phase transitions. As another useful quantity, we consider Rényi-2 Shannon entropy (SE) [37, 38] given by

$$S_S \;=\; -\log\!\left[\sum_\ell p_\ell^2\right] \;\; \text{with } p_\ell = |\langle e_\ell | \phi_0 \rangle|^2, \tag{4}$$

where the set $\{|e_\ell\rangle\}$ is a properly chosen basis set of state for $2^L$-sites spin-$1/2$ system. In the maximal decoherence limit $p_{zz} \to 1/2$, the above $S_{SE}$ can have a close relationship with $S_S$ calculated with a set of $Z$-basis for $\{|e_\ell\rangle\}$ as $\mathcal{E}^{ZZ}_j[\rho]$ in that limit is nothing but the projection operator on $Z$-domain wall states. (See Sec. VI for more details.) This fact will be utilized later on for the analysis of the numerical results.

We expect that the SEE exhibits the following system-size scaling as in the previous works [29, 30],

$$S_{SE}(L, p_{zz}) = \alpha_L L - s_0 + \mathcal{O}(L^{-1}), \tag{5}$$

where $\alpha_L$ is a non-universal coefficient depending on the ultraviolet cutoff. On the other hand, $s_0$ is a universal quantity that is independent of the system size and also ultraviolet setups, and its value is believed to be related to the low-energy properties of the system [30, 31, 36, 37]. The physical quantity $s_0$ can quantitatively capture the change in entanglement structure and the change of the mixed state, such as the emergence of strong or weak SSB. The scaling law of Eq. (5) can be also applied to the SE ($S_S$) for the critical ground state $|\phi_0\rangle$, and by setting the basis $\{|e_\ell\rangle\}$ to a set of local product states, $S_S$ corresponds to the half-cylinder "entanglement entropy" of 2D quantum Rokhsar-Kivelson (pure) wave function. There, the value of $s_0$ characterizes the (low-energy) long-range properties for the quantum state [36].

If $s_0 \neq 0$ for $\mathcal{E}(\rho_0)$ where $\rho_0$ is a pure state, the decoherence channel $\mathcal{E}[\cdot]$ is an infra-red relevant operator in the sense of renormalization group [29, 30]. In general, $e^{s_0}$ decreases if the boundary perturbation is relevant, known as "$g$-theorem" [31]. However, a recent study [30] showed that such a decreasing behavior does not necessarily hold due to dangerously irrelevant decoherence effect [30, 45].

---

[1]The detail of the definition of the strong and weak symmetries as for the $U_X$-symmetry is discussed in Appendix in [39].

# 4 Doubled Hilbert space formalism

To investigate the effect of $ZZ$-decoherence $\mathcal{E}_{tot}^{ZZ}$ on the critical ground states, we employ the doubled Hilbert space formalism [32, 33] and filtering methods [34, 35]. We shall explain these formalisms in this section.

We first consider the pure density matrix of the critical ground state of $H_{\text{TFI}}$ or $H_{\text{XXZ}}$, $\rho_0 = |\phi_0\rangle\langle\phi_0|$, and denote the original Hilbert space of the spin-1/2 system by $\mathcal{H}$. For the analysis of the decohered state $\rho_D$ through the channel $\mathcal{E}_{tot}^{ZZ}$, the doubled Hilbert space is introduced [46] as $\mathcal{H}_u \otimes \mathcal{H}_\ell$, where the subscripts $u$ and $\ell$ refer to the upper and lower Hilbert spaces corresponding to ket and bra states of mixed state density matrix, respectively. Under vectorization formula (followed by [46]) for a density matrix $\rho$ [32, 33], $\rho \longrightarrow |\rho\rangle\rangle \equiv \frac{1}{\sqrt{\dim[\rho]}} \sum_k |k^*\rangle \otimes \rho|k\rangle$, where $\{|k\rangle\}$ is an orthonormal set of states in the Hilbert space $\mathcal{H}$ and $|\rho\rangle\rangle$ resides on the doubled Hilbert space $\mathcal{H}_u \otimes \mathcal{H}_\ell$. In particular for pure state $|\phi_0\rangle$, $|\rho_0\rangle\rangle$ is given by $|\rho_0\rangle\rangle \equiv |\phi_0^*\rangle|\phi_0\rangle$, where the asterisk denotes the complex conjugation.

In this formalism, decoherence channel $\mathcal{E}[\cdot]$ is mapped to operator $\hat{\mathcal{E}}$ acting on the state vector $|\rho\rangle\rangle$ in the doubled Hilbert space $\mathcal{H}_u \otimes \mathcal{H}_\ell$ [18, 46] and denoted as $\hat{\mathcal{E}}|\rho\rangle\rangle$. Then, the decoherence channel $\mathcal{E}_{tot}^{ZZ}$ is expressed as the follows,

$$
\begin{aligned}
\hat{\mathcal{E}}_{tot}^{ZZ}(p_{zz}) &= \prod_{j=0}^{L-1}\left[ (1-p_{zz})\hat{I}_{j,u}^* \otimes \hat{I}_{j,\ell} + p_{zz}Z_{j,u}^*Z_{j+1,u}^* \otimes Z_{j,\ell}Z_{j+1,\ell} \right] \\
&= \prod_{j=0}^{L-1}(1-2p_{zz})^{1/2}e^{\tau_{zz}Z_{j,u}Z_{j+1,u}\otimes Z_{j,\ell}Z_{j+1,\ell}},
\end{aligned}
\tag{6}
$$

where $\hat{I}_{j,u(\ell)}$ is an identity operator for site-$j$ vector space in $\mathcal{H}_{u(\ell)}$, $Z(X)_{j,u(\ell)}$ is Pauli-$Z(X)$ operator at site $j$ in the space $\mathcal{H}_{u(\ell)}$ and $\tau_{zz} = \tanh^{-1}[p_{zz}/(1-p_{zz})]$. The application of the channel operator $\hat{\mathcal{E}}_{tot}^{ZZ}(p_{zz})$ changes the initial state $|\rho_0\rangle\rangle$ and also the norm of the vector $|\rho_0\rangle\rangle$. In this sense, the operation $\hat{\mathcal{E}}_{tot}^{ZZ}(p_{zz})$ is non-unitary. Here, note that the initial doubled state $|\rho_0\rangle\rangle$ is nothing but the ground state of the two decoupled TFIM or XXZ model on a *two-leg spin-1/2 ladder* with the Hilbert space $\mathcal{H}_u \otimes \mathcal{H}_\ell$ and the Hamiltonian $H_{\text{TFI(XXZ)}}^u + H_{\text{TFI(XXZ)}}^\ell$ on the upper and lower chains.

In the ladder system, the decohered state $|\rho_D\rangle\rangle$ is given as

$$
|\rho_D\rangle\rangle \equiv \hat{\mathcal{E}}_{tot}^{ZZ}|\rho_0\rangle\rangle = C(p_{zz}, L)\prod_{j=0}^{L-1}\left[ e^{\tau_{zz}\hat{h}_{j,j+1}^{zz}} \right]|\rho_0\rangle\rangle,
\tag{7}
$$

where the operators $\hat{h}_{j,j+1}^{zz} = Z_{j,u}Z_{j+1,u} \otimes Z_{j,\ell}Z_{j+1,\ell}$ and $C(p_{zz}, L) \equiv (1-2p_{zz})^{L/2}$. The filtering is the application of the operator $\hat{\mathcal{E}}_{tot}^{ZZ}$ to the state $|\rho_0\rangle\rangle$. This operation can be regarded as a non-unitary imaginary-time evolution where the time interval is $\tau_{zz}(p_{zz})$. We investigate the properties of the mixed $\rho_D$ by studying its counterpart $|\rho_D\rangle\rangle$. In particular, since the norm $\langle\langle\rho_D|\rho_D\rangle\rangle$ corresponds to the purity $\text{Tr}[\rho_D^2]\,(>0)$, the SEE, which is one of the target quantities in this work, is given by the logarithm of the norm [30]

$$
S_{SE}(p_{zz}, L) = -\log\langle\langle\rho_D|\rho_D\rangle\rangle.
\tag{8}
$$

Before numerically observing the effect of $ZZ$-decoherence on the critical ground states, we expect that for small $p_{zz}$, the $ZZ$-decoherence is irrelevant and the initial critical properties are preserved. On the other hand, for $p_{zz} \to 1/2$ ($\lambda_{zz} \to \infty$), the decoherence $ZZZZ$ effect gives a significant impact to the ground states: The decohered mixed states can have a long-range order (LRO) such as $Z_2$ SWSSB. To examine this issue is one of the main subjects of

the present study. Obviously, an emergent state with a LRO is distinct from the initial critical state $|\rho_0\rangle\rangle$. (2-1) For large $p_{zz}$, it is naturally expected that the LRO ($Z_2$ SWSSB) generates cat state property ($Z_2$ ensemble pair) in the density matrix. Then, a double-critical mixed state emerges, leading to a possibility that the SEE is twice that with a single-site $Z$ decoherence studied in [30]. We later verify these expectations by observing the SEE and other physical quantities.

## 5   Numerical analysis of system environmental entanglement

We numerically study the decohered state $|\rho_D\rangle\rangle$ by using the MPS to analyze large systems and to calculate SEE and some correlators characterizing orders such as SWSSB emerging in the decohered state vector $|\rho_D\rangle\rangle$. We prepare the initial critical state $|\rho_0\rangle\rangle$ by using DMRG in the TeNPy package [47, 48]. The filtering operation $\hat{\mathcal{E}}_{tot}^{ZZ}(p_{zz})$ in Eq. (6) applied to the MPS $|\rho_0\rangle\rangle$ can be efficiently carried out by making use of the libraries in TeNPy [47, 48].

(1-3)(2-3) Here, we explain details of the numerical simulations performed in this work. In the DMRG simulations for preparing initial ground states, we set maximum bond dimension $D = 240$-$300$ and truncate the singular value less than $\mathcal{O}(10^{-7})$. In this set up,the energy convergence of the iterative DMRG sweep is $\Delta E < \mathcal{O}(10^{-6})$ for the initial MPS ground state. In the filtering operation through the matrix product operators, the cutoff of singular values used in the singular-value decomposition is set $\mathcal{O}(10^{-10}-10^{-15})$ for sufficiently precise calculations. Under this setup, we take the system size up to $L = 28$ ladder (the total bonds are $3L$).

In the numerical calculation, we also introduce the (reduced) susceptibility of Rényi-2 correlator to corroborate the observation of the properties of the mixed state $\rho_D$. The susceptibility is given by

$$\chi_{ZZ}^{\mathrm{II}} = \frac{2}{L} \sum_{r=1}^{L/2} C_{ZZ}^{\mathrm{II}}(0, r),$$

where $C_{ZZ}^{\mathrm{II}}$ is the Rényi-2 correlator for the state $|\rho_D\rangle\rangle$,

$$C_{ZZ}^{\mathrm{II}}(i, j) \equiv \frac{\langle\langle \rho_D | Z_{i,u} Z_{j,u} Z_{i,\ell} Z_{j,\ell} | \rho_D \rangle\rangle}{\langle\langle \rho_D | \rho_D \rangle\rangle}.$$

In the original 1D system, a counterpart of $C_{ZZ}^{\mathrm{II}}(i, j)$ is given by

$$C_{ZZ}^{\mathrm{II}}(i, j) \equiv \frac{\mathrm{Tr}[Z_i Z_j \rho_D Z_j Z_i \rho_D]}{\mathrm{Tr}[(\rho_D)^2]}.$$

This observable provides us an order parameter for detecting both the LRO and SSB of the strong symmetry, but *not* those of the weak symmetry [18–20, 49]. Note that this quantity $C_{ZZ}^{\mathrm{II}}(i, j)$ exhibits non-trivial behavior in the whole parameter region, that is, it varies concomitantly with the decoherence nature of the state.

In general, one can consider another quantity, the canonical correlation function, given by

$$C_Z^{\mathrm{I}}(i, j) = \mathrm{Tr}[\rho_D Z_i Z_j] = \frac{\langle\langle \mathbf{1} | Z_{i,u} Z_{j,u} | \rho_D \rangle\rangle}{\langle\langle \mathbf{1} | \rho_D \rangle\rangle},$$

where $|\mathbf{1}\rangle\rangle \equiv \frac{1}{2^{3L/2}} \prod_{j=0}^{L-1} |t\rangle_j$ with $|t\rangle_j = |\uparrow_u\uparrow_\ell\rangle_j + |\downarrow_u\downarrow_\ell\rangle_j$. The observable $C_Z^{\mathrm{I}}(i, j)$ can be an order parameter characterizing the genuine SSB (i.e., weak symmetry SSB) [18–20]. However, in our target system subject to the $ZZ$-decoherence, this quantity is invariant under the

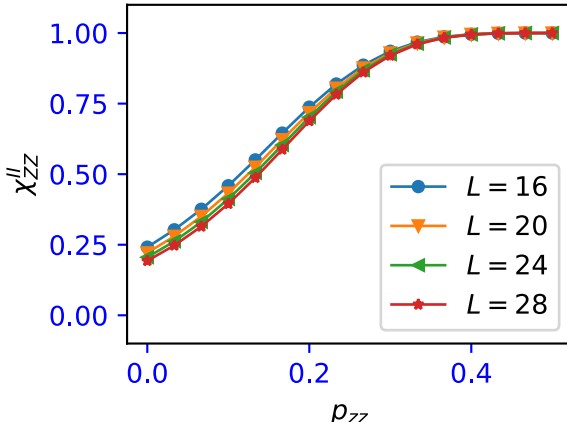

Figure 1: Behaviors of $\chi_{ZZ}^{\mathrm{II}}$ for TFIM critical states under $ZZ$-decoherence. Data for various system sizes are displayed, indicating negligibly small system-size dependence.

decoherence since $\mathrm{Tr}[\rho_D Z_i Z_j] = \mathrm{Tr}[\mathcal{E}_{tot}^{ZZ}(\rho_0) Z_i Z_j] = \mathrm{Tr}[\rho_0 Z_i Z_j]$, and $C_Z^{\mathrm{I}}(i,j)$ keeps the value of the pure critical state of $\rho_0$ for any $p_{zz}$. Thus, $|C_Z^{\mathrm{I}}(i,j)|$ has a power law decay, $\propto \frac{1}{|i-j|^\eta}$ as a function of $|i-j|$. The power law decay corresponds to that of $c = 1/2$ and $c = 1$ CFT for the TFIM and XXZ model, respectively.

Before showing the numerical calculations, we discuss the diagnosis of non-trivial mixed states from the viewpoint of the above two quantities, $C_{ZZ}^{\mathrm{II}}(i,j)$ and $|C_Z^{\mathrm{I}}(i,j)|$.

The combination of $C_{ZZ}^{\mathrm{II}}(i,j)$ and $|C_Z^{\mathrm{I}}(i,j)|$ can detect various symmetry breaking phases including the SWSSB, which is recently proposed in Refs. [18–20]. In the system with the strong $Z_2$ symmetry [2], if a state exhibits $C_{ZZ}^{\mathrm{II}}(i,j) \sim \mathcal{O}(1)$ and $|C_Z^{\mathrm{I}}(i,j)| \sim 0$ for $|i-j| \to \infty$, SSB of the off-diagonal (i.e., strong) symmetry occurs, and the diagonal (i.e., weak) symmetry is preserved [18]. This is SWSSB. Also if $C_{ZZ}^{\mathrm{II}}(i,j) \sim \mathcal{O}(1)$ and $|C_Z^{\mathrm{I}}(i,j)| \sim \mathcal{O}(1)$, then both the weak and strong SSBs occur called strong-to-trivial SSB. [If the reader is interested in a brief explanation of strong and weak symmetries as well as their combination of SSB and the notion of SWSSB, see [16, 39].]

Since $|C_Z^{\mathrm{I}}(i,j)|$ of the systems exhibits the power-law decay for any decoherence strength, $|C_Z^{\mathrm{I}}(i,j)| \sim 0$ for $|i-j| \to \infty$ limit, the target mixed state exhibits the SWSSB or remains symmetric as decoherence gets strong.

## 5.1 Numerical results for critical TFIM

We numerically observe effects of $ZZ$-decoherence on the critical state of the TFIM. First, calculations of the observables $\chi_{ZZ}^{\mathrm{II}}$ is shown in Fig. 1. As shown in Fig. 1 (a), $\chi_{ZZ}^{\mathrm{II}}$ increases with $p_{zz}$ and saturates to unity for large $p_{zz}$. We observe the existence of the plateau regime with $\chi_{ZZ}^{\mathrm{II}} = 1$ for $p_{zz} \gtrsim 0.4$, without system size dependence, while for $p_{zz} \lesssim 0.3$, small system size dependence appears, where the value decreases as increasing system size $L$. In particular, for the limit $p_{zz} = 0$, we carefully verified that $\chi_{ZZ}^{\mathrm{II}}$ exhibits a power-law-decreasing behavior with respect to $L$ [not shown] and its power exponent is fairly close to the value predicted by the conventional correlation function of the critical ground state of the TFIM, and for $L \to \infty$, $\chi_{ZZ}^{\mathrm{II}}$ smoothly approaches zero with that exponent. This behavior implies that in our finite-

---

[2]Strictly, to define the SWSSB, we require that the initial state, target, decoherence channel, and final decohered state satisfy to be strongly-symmetric for a target on-site symmetry [19, 20].

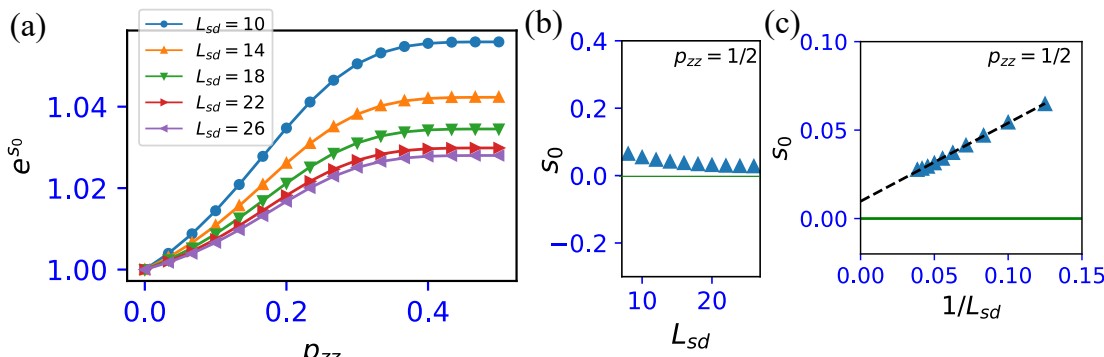

Figure 2: (a)$p_{zz}$-dependence of the $g$-function $e^{s_0}$ for various sets of system size $\{L_{sd}, L_{sd}+2, L_{sd}+4, L_{sd}+6\}$ for critical TFIM. (b) $L_{sd}$-dependence of the extracted value of $s_0$ for $p_{zz} = 1/2$. The green line, $-\gamma_{n=2} + \log 2 = -\log 2 + \log 2 = 0$, where $\gamma_{n=2}$ is the estimation value of the subleading term of the Rényi-2 Shannon entropy in the previous work [37,38], the value of which is $\gamma_{n=2} = \log 2$ [38]. The values of $s_0$ was extracted by the numerical fitting procedure by using the set of four different system sizes, $\{L_{sd}, L_{sd}+2, L_{sd}+4, L_{sd}+6\}$. (c) The extrapolation of $s_0$ for $p_{zz} = 1/2$ in $L_{sd} \to \infty$. The linear fitting function is estimated as $s_0 = -0.44(1/L_{sd}) + 0.01$.

size numerics, a sharp phase transition does not occur and the state smoothly approaches the unique state for $p_{zz} \to 1/2$.

As mentioned in the previous section, $|C_Z^{\mathrm{I}}(i,j)| \sim 0$ for $|i-j| \to \infty$, the numerical result indicates that the state for $p_{zz} > 0.4$ is in a SWSSB phase. That is, large $ZZ$-decoherence induces the transfer of the system from the symmetric phase to the SWSSB phase, although a clear phase transition point cannot be identified.

We next focus on the SEE. Here, we observe how $g$-function $e^{s_0}$, extracted from the various system size data, behaves as $p_{zz}$ increases. The details of the practical procedure and a concrete fitting example are shown in Appendix A. Results are shown in Fig. 2, and we find that the SEE is well-fitted by the scaling law Eq. (5). As shown in Fig. 2 (a), the $g$-function $e^{s_0}$ increases slightly with $p_{zz}$. We further observe that the different system size data do not cross with each other, and then, $e^{s_0}$ does not indicate the existence of a phase transition induced by the $ZZ$-decoherence. Combined with the results of $\chi_{ZZ}^{\mathrm{II}}$, the $ZZ$-decoherence simply induces a crossover from the critical state of the TFIM to the mixed state with critical properties. Furthermore, interestingly enough, Fig. 2 (b) shows that the values of $s_0$ at $p_{zz} = 1/2$ (the exponent of the extracted $g$-function) for the various system sizes are very close to the value obtained from the universal constant $\gamma_{n=2} = \log 2$ in the SE measured by the single-site $Z$-basis [38], as $-\gamma_{n=2} + \log 2 = 0$. In order to verify this observation, we perform the extrapolation of $s_0$ for $L_{sd} \to \infty$, and find that $s_0$ approaches zero as observed in Fig. 2 (c). Detailed discussion on this point, especially the reason why the obtained $s_0$ takes $-\gamma_{n=2} + \log 2 = 0$, is given later on.

In Appendix C, we show how the critical ground state of the TFIM evolves under $X + ZZ$-decoherence. In the previous paper [39], we investigated this system from the viewpoint of SWSSB, and obtained interesting results showing the existence of a phase transition at a finite strength of decoherence. Numerical results of the $g$-function in Appendix C exhibit a similar phase transition behavior. Then, an important and interesting question is how these two observables relate to each other. This issue is discussed in detail in subsequent sections for the XXZ model, which exhibits similar behaviors with the TFIM under the $X + ZZ$-decoherence.

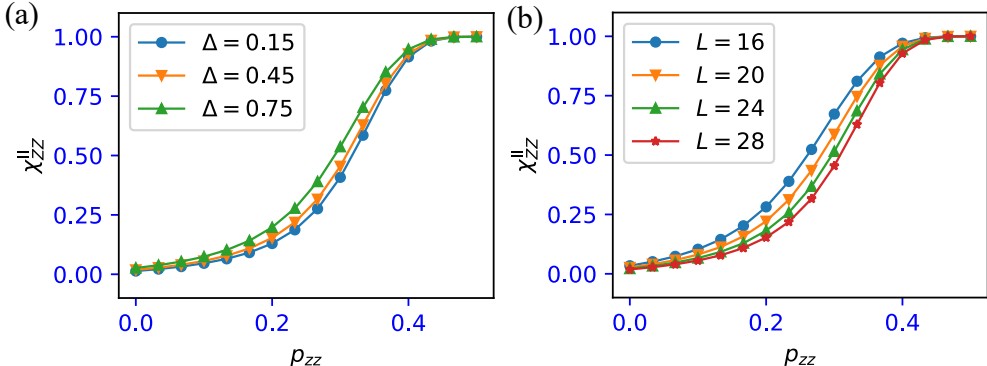

Figure 3: (a) Behaviors of $\chi_{ZZ}^{\mathrm{II}}$ for XXZ critical states under $ZZ$-decoherence. (b) System size dependence of $\chi_{ZZ}^{\mathrm{II}}$ for $\Delta = 0.45$. We set $L = 28$ (total 56 sites).

## 5.2 Numerical results for critical XXZ model

We turn to the numerical study on the critical ground state of the XXZ model. The $p_{zz}$-dependence of $\chi_{ZZ}^{\mathrm{II}}$ for various values of $\Delta$ (in the TLL regime) is shown in Figs. 3 (a). As seen in Fig. 3 (a), $\chi_{ZZ}^{\mathrm{II}}$ increases with $p_{zz}$ and saturates for large $p_{zz}$, that is, the plateau regime with $\chi_{ZZ}^{\mathrm{II}} = 1$ emerges in $p_{zz} \gtrsim 0.4$. We further observe the system size dependence with $\Delta$ fixed as shown in Fig. 3 (b). The small but finite system size dependence of $\chi_{ZZ}^{\mathrm{II}}$ exists in the intermediate $p_{zz}$ regime, but for $p_{zz} \gtrsim 0.4$, $\chi_{ZZ}^{\mathrm{II}}$ has a finite value without system size dependence. As mentioned in the previous section, $|C_Z^{\mathrm{I}}(i,j)|$ exhibits power law decay as a function of the distance $|i - j|$ and $|C_Z^{\mathrm{I}}(i,j)| \sim 0$ for $|i - j| \to \infty$. Thus, for $p_{zz} \gtrsim 0.4$ regime, the SWSSB mixed phase is expected to emerge for any value of $\Delta$. Here, we ask if there exists a sharp phase transition to the SWSSB state. While estimation of critical decoherence to the SWSSB is not so easily from the data of $\chi_{ZZ}^{\mathrm{II}}$, its existence is not denied from the calculation of $g$-function in Fig. 4 as we discuss below.

Next, we move on to the calculation of the SEE. Here, we observe how $g$-function $e^{s_0}$, extracted from the various system size data, behaves in the region where the mixed state changes into the SWSSB state as $p_{zz}$ increases.

We calculate the SEE and confirm that the SEE is well-fitted by the scaling law of Eq. (5) and extract the $g$-function. Then, we observe the $g$-function for $\Delta = 0.45$ varying the system size. The results are displayed in Fig. 4 (a). The $g$-function $e^{s_0}$ increases as $p_{zz}$ increases, and we find that all system-size data lines cross with each other at $p_{zz} \sim 0.4$. This value of $p_{zz}$ coincides with the saturation point of $\chi_{ZZ}^{\mathrm{II}}$ in Fig. 3(a). This behavior is observed for other $\Delta$'s, as shown in Appendix B. This crossing of the data indicates the existence of the phase transition between the critical and the mixed state with strong Rényi-2 correlation.

To elucidate the properties of the transition, we perform a finite-size scaling analysis for the $g$-function by employing the most general form of scaling ansatz,

$$e^{s_0} = L_{sd}^{\zeta/\nu} g((p_{zz} - p_{zz}^c)L_{sd}^{1/\nu}),$$

where $p_{zz}^c$ is the critical transition point and $\zeta$ and $\nu$ are critical exponents. The scaling analysis was carried out with the help of pyfssa [50, 51]. The result is shown in Fig. 4(b), where the clear data collapse is observed, supporting the genuine phase transition. There, we estimate $p_{zz}^c = 0.439(0)$ with $\zeta = 0.007(3)$ and $\nu = 2.519(8)$. The value of $\zeta$ is close to zero, similar to the scaling-analysis result performed in the previous works [29, 36]. These numerical results indicate the existence of a phase transition between the critical XXZ state and the mixed state

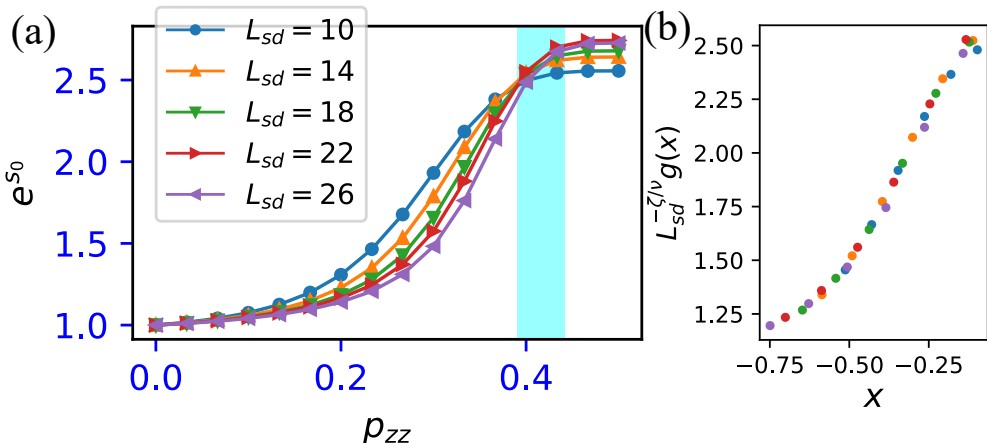

Figure 4: (a) $p_{zz}$-dependence of the $g$-function $e^{s_0}$ for various sets of system size $\{L_{sd}, L_{sd}+2, L_{sd}+4, L_{sd}+6\}$ for critical XXZ model. The values of $s_0$ was extracted by the numerical fitting procedure by using the set of four different system size, $\{L_{sd}, L_{sd}+2, L_{sd}+4, L_{sd}+6\}$. We set $\Delta = 0.45$. (b) Scaling data collapse, where the label $x = (p_{zz} - p_{zz}^c)L_{sd}^{1/\nu}$. We used data points within $0.2 \leq p_{zz} \leq 0.5$. We estimated $p_c = 0.439(0)$ with $\zeta = 0.007(3)$ and $\nu = 2.519(8)$.

with strong Rényi-2 correlation(SWSSB) [3]

Next, we show the most interesting numerical results in this work. We observe $\Delta$-dependence of the $g$-function for $p_{zz} = \frac{1}{2}$. The results are displayed in Fig. 5. Surprisingly enough, we find the value of $e^{s_0}$ is very close to $2\sqrt{2K}$, where the value of $\sqrt{2K}$ was estimated by analytical methods and verified numerically in the previous study on the XXZ chain *under single-site Z-decoherence* [30]. This multiple factor "2" discrepancy can be related to the long-range nature of the Rényi-2 correlation in the decohered mixed state. In the thermodynamic limit, the $Z_2$ strong symmetry is spontaneously broken, and it is realized at the ensemble level. However, in the finite system, a GHZ (cat) state emerges respecting the $Z_2$ strong symmetry, and this long-range entanglement can be the origin of the multiple factor "2".

We also find that the origin of the multiple factor "2" is understood by analytically observing the connection of Rényi-2 Shannon entropy for $p_{zz} = 1/2$ limit. The numerical assisted analytical understanding is shown in the following section. By this observation, it is clarified that emergence of the GHZ state is an essential ingredient of this phenomenon.

In addition, we numerically investigate effects of another decoherence for the critical state of the TFIM, and we find that the $g$-function behaves non-trivially for strong decoherence. This result is shown in Appendix B.

---

[3]Obviously whether this transition observed by the spin correlations and that of system-environment entanglement observed by $e^{s_0}$ take place simultaneously is an interesting question. A plausible possibility is that there exists a single phase transition between the critical state and decohered mixed state, but different observables exhibit its signal at different values of $p_{zz}$. A typical example is the KT transition of the 2D classical XY spin model, where the specific heat and correlation functions give different values of the transition [52]. For the present systems, it is a future problem.

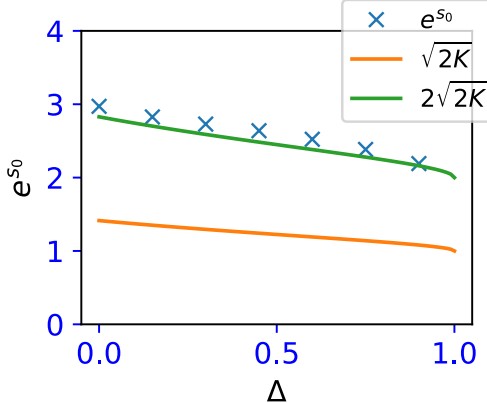

Figure 5: $\Delta$-dependence of $g$-function $e^{s_0}$ for $p_{zz} = 1/2$ limit. The values of $s_0$ for each $\Delta$ were extracted by the numerical fitting procedure for the set of the different system sizes $L = 12, 16, 20, 24, 28$ data. The orange and green lines are $e^{s_0} = \sqrt{2K(\Delta)}$ and $e^{s_0} = 2\sqrt{2K(\Delta)}$, respectively.

## 6 Analysis of universal $s_0$ for $p_{zz} = 1/2$ limit

In the previous section, we numerically observed the universal term $s_0$. The values of $e^{s_0}$ change from one to nontrivial values for both critical spin systems. In particular, for the decoherence limit $p_{zz} = 1/2$, we found that the values of $e^{s_0}$ are related to the previously studied ones in [30, 36]. In this section, we discuss this relationship by analytically studying the connection between the SEE for $\rho_D$ with $p_{zz} = 1/2$ limit and the Rényi-2 Shannon entropy for the glassy GHZ basis. Numerical study is also used to corroborate the observation.

### 6.1 Glassy GHZ expansion of $\rho_D$ for $p_{zz} = 1/2$ decohered limit

We first prove that the decohered state $\rho_D$ for $p_{zz} = 1/2$ limit (projective $ZZ$-measurement limit) can be expanded by the glassy GHZ states, that is, we find the following representation: For $p_{zz} = 1/2$ $ZZ$-decoherence limit, the state $\rho_D$ is expanded as

$$\mathcal{E}_{tot}^{ZZ}[\rho_0]_{p_{zz}=1/2} = \sum_{g,\alpha=\pm} P^{(g,\alpha)} \rho_0 P^{(g,\alpha)}, \tag{9}$$

where $P^{(g,\alpha)} = |g^\alpha\rangle\langle g^\alpha|$, and $\{|g^\alpha\rangle\}$ are a set of the glassy GHZ states of $L$-site spin system. The glassy GHZ basis is labeled by the number $g = 0, 1, \cdots, 2^{L-1} - 1$ and $\alpha$ labels the parity for the global spin flip $\mathbf{Z}_2$ operator $U_X$ and $U_X|g^\pm\rangle = \pm|g^\pm\rangle$.
For example, $|g^\pm = 1^\pm\rangle = \frac{1}{\sqrt{2}}[|\uparrow\downarrow\uparrow\cdots\uparrow\rangle \pm |\downarrow\uparrow\downarrow\cdots\downarrow\rangle]$.

Here, we comment that Eq. (9) is satisfied for any $\rho_0$ if $\rho_0$ is an eigenstate of $U_X$, that is, $\rho_0$ is given as $\rho_0 = |\phi_0\rangle\langle\phi_0|$ and $|\phi_0\rangle$ is symmetric under $U_X$ in the present case.

We shall prove Eq. (9) in the following way;
First, $\mathcal{E}_{tot}^{ZZ}[\rho]_{p_{zz}=1/2}(\equiv \rho_{D,p_{zz}=1/2})$ can be rewritten as

$$\rho_{D,p_{zz}=1/2} = \sum_{\vec{\beta}} \left( P_{L-1}^{ZZ,\beta_{L-1}} P_{L-2}^{ZZ,\beta_{L-2}} \cdots P_1^{ZZ,\beta_1} P_0^{ZZ,\beta_0} \right) \rho_0 \left( P_0^{ZZ,\beta_0} P_1^{ZZ,\beta_1} \cdots P_{L-2}^{ZZ,\beta_{L-2}} P_{L-1}^{ZZ,\beta_{L-1}} \right), \tag{10}$$

where $P_j^{ZZ,\beta_j}$ is the projection operator defined by $P_j^{ZZ,\beta_j} = \frac{1+\beta_j Z_j Z_{j+1}}{2}$ with the outcome $\beta_j$ taking $\pm 1$, and $\vec{\beta} = \{\beta_0, \beta_1, \cdots, \beta_{L-2}, \beta_{L-1}\}$. Note that there is a crucial constraint for the outcome, that is, $\beta_{L-1}$ is fixed by the patterns of $\{\beta_0, \beta_1, \cdots, \beta_{L-2}\}$, $\beta_{L-1} = \prod_{j=0}^{L-2} \beta_j$, coming from the constraint for $ZZ$-measurement operator, $\prod_{j=0}^{L-1} Z_j Z_{j+1} = 1$. The sum $\sum_{\vec{\beta}}$ in Eq. (10), therefore, means the summation of total $2^{L-1}$ outcome patterns of $\{\beta_0, \beta_1, \cdots, \beta_{L-2}\}$.

By making use of the identity $1 = \sum_c |c\rangle\langle c|$, where $|c\rangle$ is a $L$-site basis state of the local spin $Z_j$ such as $|\uparrow\uparrow\uparrow \cdots \uparrow\rangle$ ($\sum_c$ means all sum of product $Z$ basis pattern.), Eq. (10) can be expressed as,

$$
\begin{aligned}
[\text{Eq. (10)}] &= \sum_{\vec{\beta}} \left( P_{L-1}^{ZZ,\beta_{L-1}} P_{L-2}^{ZZ,\beta_{L-2}} \cdots P_1^{ZZ,\beta_1} P_0^{ZZ,\beta_0} \right) \left[ \sum_c |c\rangle\langle c| \right] \rho_0 \left[ \sum_c |c\rangle\langle c| \right] \\
&\quad \times \left( P_0^{ZZ,\beta_0} P_1^{ZZ,\beta_1} \cdots P_{L-2}^{ZZ,\beta_{L-2}} P_{L-1}^{ZZ,\beta_{L-1}} \right) \\
&= \sum_{c-\{\bar{c}\}} \left[ |c\rangle\langle c| + |\bar{c}\rangle\langle \bar{c}| \right] \rho_0 \left[ |c\rangle\langle c| + |\bar{c}\rangle\langle \bar{c}| \right] \\
&= \sum_{c-\{\bar{c}\}} \left[ K_c \rho_0 K_c + K_c \rho_0 N_c + N_c \rho_0 K_c + N_c \rho_0 N_c \right],
\end{aligned}
\tag{11}
$$

where we have introduced projective operators defined as $K_c = |c\rangle\langle c|$ and $N_c = |\bar{c}\rangle\langle \bar{c}|$, and note that $|c\rangle$ and $|\bar{c}\rangle$ are a "parity pair" given by $|c\rangle = U_X |\bar{c}\rangle$. The sum $\sum_{c-\{\bar{c}\}}$ denotes the summation over the subset of basis $\{|c\rangle\}$ (the total element of the subset is $2^{L-1}$), that is, each elements of which are not connected by the parity $U_X$.

On the other hand, we note that since the glassy GHZ basis can be written by $|g^\pm\rangle = \frac{1}{\sqrt{2}}[|c\rangle \pm |\bar{c}\rangle]$, the RHS of Eq. (9) can be expanded as

$$
\begin{aligned}
[\text{RHS of Eq. (9)}] &= \sum_{c-\{\bar{c}\}} \frac{1}{2} \Big[ K_c \rho_0 K_c + K_c \rho_0 N_c + L_c \rho_0 L_c + L_c \rho_0 M_c \\
&\quad + M_c \rho_0 L_c + M_c \rho_0 M_c + N_c \rho_0 K_c + N_c \rho_0 N_c \Big],
\end{aligned}
\tag{12}
$$

where $L_c = |c\rangle\langle \bar{c}|$ and $M_c = |\bar{c}\rangle\langle c|$.

Then, if the state $\rho_0$ is unique critical ground state of the target Hamiltonian, $U_X |\phi_0\rangle = \pm|\phi_0\rangle$ since $U_X$ commutes with the Hamiltonian and $(U_X)^2 = 1$. This fact directly leads to

$$
\langle c|\phi_0\rangle = \pm\langle \bar{c}|\phi_0\rangle.
\tag{13}
$$

By substituting the above relation into Eq. (12), we verify that Eq. (12) is equal to Eq. (11), regardless of the sign on the RHS of Eq. (13). Thus, Eq. (9) has been proved.

In general, the sign of Eq.(13) can depend on the system size, boundary conditions and the model parameters. To further validate the above argument, the sign of Eq.(13) is observed for practical cases by using the exact diagonalization to find that the following relations hold for the critical states in both the TFIM and XXZ models,

$$
\langle c|\phi_0\rangle = \pm\langle \bar{c}|\phi_0\rangle \text{ for XXZ critical,}
\tag{14}
$$

$$
\langle c|\phi_0\rangle = \langle \bar{c}|\phi_0\rangle \text{ for TFIM critical.}
\tag{15}
$$

Here, we comment that for the above XXZ case, numerically, the sign depends only on the system size and there is no $\Delta$-dependence.

## 6.2   SEE-SE correspondence for $ZZ$-projective measurement limit

Equation (9) gives an important relation: the SEE for $\rho_{D,p_{zz}=1/2}$ denoted by $S_{SE,p_{zz}=1/2}$ is written as

$$S_{SE,p_{zz}=1/2} = -\log \text{Tr}[\rho_{D,p_{zz}=1/2}^2] \overset{\text{Eq. (9)}}{=} -\log\Bigg[\sum_{g,\alpha=\pm} |\langle g^\alpha|\phi_0\rangle|^4\Bigg]. \tag{16}$$

The last quantity in Eq. (16) can be regarded as the Rényi-2 Shannon entropy $S_S$ [37, 38] of the critical state $|\phi_0\rangle$ *in terms of the glassy GHZ basis* $\{|g^\alpha\rangle\}$.

As a result, we find that the SEE of the mixed state $\rho_D$ for $p_{zz} = 1/2$ limit corresponds to the Rényi-2 Shannon entropy. This observation sheds light on the results of $s_0$ and its $g$-function $e^{s_0}$ obtained in the previous section, as we explain in the following subsection.

## 6.3   Relation between SEE of Z-decoherence limit and SEE of $ZZ$-decoherence limit

We further find an interesting relation between the SEE for $p_{zz} = 1/2$ limit $S_{SE,p_{zz}=1/2}$ and the Rényi-2 Shannon entropy of the critical state $|\phi_0\rangle$ in terms of $Z$-product basis. This corresponds to the case, in which on-site local maximal $Z_j$-decoherence is applied to the critical state $|\phi_0\rangle$ at all system sites, previously studied in [30, 36]. In this case,

$$\begin{aligned}
\text{Tr}[\rho_{D,p_{zz}=1/2}^2] &= \text{Tr}\Bigg[\sum_{c-\{\bar{c}\}} (K_c\rho_0 K_c\rho_0 K_c + K_c\rho_0 L_c\rho_0 K_c + L_c\rho_0 K_c\rho_0 L_c + L_c\rho_0 L_c\rho_0 L_c)\Bigg] \\
&= \sum_c |\langle c|\rho_0|c\rangle|^2 + \sum_c |\langle c|\rho_0|\bar{c}\rangle|^2.
\end{aligned} \tag{17}$$

Then, by using the numerical observation of Eq. (14) or Eq. (15), we easily obtain

$$\text{Tr}[\rho_{D,p_{zz}=1/2}^2] = 2\sum_c |\langle c|\rho_0|c\rangle|^2. \tag{18}$$

The above equation leads to the following relation

$$S_{SE,p_{zz}=1/2} = -\log\Bigg[\text{Tr}[\rho_{D,p_{zz}=1/2}^2]\Bigg] = -\log\Bigg[2\sum_c |\langle c|\rho_0|c\rangle|^2\Bigg] = S_S(\{|c\rangle\}) - \log 2. \tag{19}$$

That is, the SEE for $p_{zz} = 1/2$ limit relates to the Rényi-2 Shannon entropy of the critical state $|\phi_0\rangle$ in terms of $Z$-product basis (denoted by $S_S(\{|c\rangle\})$) with the deviation "*minus* $\log 2$".

Then, since the Rényi-2 Shannon entropy of the critical state $|\phi_0\rangle$ in terms of $Z$-product basis $S_S(\{|c\rangle\})$ corresponds to the SEE for the critical state under the on-site local $Z_j$-decoherence limit [30], denoted by $S_{SE,p_z=1/2}^Z$. Thus, $S_{SE,p_{zz}=1/2}$ relates to $S_{SE,p_z=1/2}^Z$ with the deviation "*minus* $\log 2$", where the precise form of $S_{SE,p_z=1/2}^Z$ is already known [30, 37, 38].

Finally, we get a scaling law of $S_{SE,p_{zz}=1/2}$ and its universal term $s_0$ for both TFIM and XXZ models by making use of the previous studies [30, 37, 38], in which the scaling law of $S_{SE,p_z=1/2}^Z$ and values of the universal term $s_0$ are already obtained as $S_{SE,p_z=1/2}^Z = \alpha_1^{\text{TFIM(XXZ)}}L - s_0^{\text{TFIM(XXZ)}} + \mathcal{O}(L^{-1})$. Then, $S_{SE,p_{zz}=1/2}$ has the following forms:

$$S_{SE,p_{zz}=1/2} = \begin{cases} \alpha_1^{\text{TFIM}}L - (s_0^{\text{TFIM}} + \log 2) + \mathcal{O}(L^{-1}) & \text{for TFIM critical} \\ \alpha_1^{\text{XXZ}}L - (s_0^{\text{XXZ}} + \log 2) + \mathcal{O}(L^{-1}) & \text{for XXZ critical,} \end{cases}$$

where $\alpha_1^{\text{TFI(XXZ)}}$ are non-universal coefficients. Also, it is known that $s_0^{\text{TFIM}} = -\log 2$ [37, 38]. The $g$-function of the universal parts for $S_{SE,p_{zz}=1/2}$, $e^{s_0}$ are regarded as

$$e^{s_0} = \begin{cases} e^0 & \text{for TFIM and from the result in [37, 38]} \\ 2\sqrt{2K} & \text{for XXZ and from the result in [30].} \end{cases}$$

(20)

In particular, we find that the $g$-function $e^{s_0}$ of $S_{SE,p_{zz}=1/2}$ for the critical XXZ model has a multiple factor "2" compared to the $g$-function $e^{s_0}$ of $S_{SE,p_z=1/2}^Z$ in Ref. [30], which is consistent to the result in Fig. 5.

Then, the numerical results of the $g$-function for $p_{zz} = 1/2$ limit of the critical TFIM shown in Fig. 2 are consistent with the TFIM result of Eq. (20).

# 7   Conclusion

(1-2)In this study, we investigated how decoherence alters the critical pure states by using the concrete example—the two-site nearest-neighbor decoherence channel. Compared to the decoherence of on-site operator, the critical pure state can change to a mixed state with rich non-trivial orders specific to the mixed state. Concretely, this work considered a simple nearest-neighbor decoherence channel, i.e., the $ZZ$-decoherence that acts on the critical states for the TFIM and XXZ models. By making use of the DMRG and filtering methods as numerical methods, we investigated how the SEE behaves and how criticality changes due to the effects of the $ZZ$-decoherence. For both the TFIM and XXZ models, we found that the SEE exhibits the scaling law $\alpha L - s_0 + \mathcal{O}(L^{-1})$ for any strength of the $ZZ$-decoherence, and that the $g$-function, $e^{s_0}$, changes its value as varying the strength of decoherence. Simultaneously, the decoherence induces the change in the mixed states where a long-range-ordered state appears as observed by the Rényi-2 correlator. We especially emphasize that the $ZZ$-decoherence induces a drastic change of the mixed state in the XXZ model, whereas for the critical TFIM, it does not. As a summary, we here comment on our findings for the $g$-factor:

(I) For the TFIM, the value of the $e^{s_0}$ continuously changes and approaches unity ($s_0$ approaches zero) as the decoherence increases. We numerically demonstrated that the estimated value $s_0 = 0$ for $p_{zz} = 1/2$ limit is related to the subleading term of the Rényi-2 Shannon entropy in the previous work [37, 38].

(II) The case of the XXZ model has rich physical phenomena. In particular, the value of $e^{s_0}$ for the critical XXZ model under strong $ZZ$-decoherence is twice that of the previous study on $Z$-decoherence obtained by the CFT and RG analysis [30]. Our numerical findings of the value of $e^{s_0}$ were analytically understood. For the $p_{zz} = 1/2$ limit, the SEE in our system corresponds to the Rényi-2 Shannon entropy for the glassy GHZ set of basis, value of which is related to the Rényi-2 Shannon entropy for $Z$-product basis. Therefore, our numerically estimated $e^{s_0}$'s are related to the values of $e^{s_0}$ obtained in the previous studies [30, 37, 38].

(1-2)From these findings, we found that, as observed in the XXZ model, the $ZZ$-decoherence preserving the $Z_2$ symmetry does not destroy the initial pure-state criticality; instead, it enhances the SWSSB order and generates a new nonunitary critical point. The universality is elucidated by the SEE and its $g$-factor. From the numerical findings of the XXZ model, we expect that the value of $e^{s_0}$ reflects a symmetry-breaking pattern inherent in the decoherence channel. In other words, it originates from GHZ-type long-range quantum correlations formed between the system and the environment. Such a structure has no counterpart in pure states.

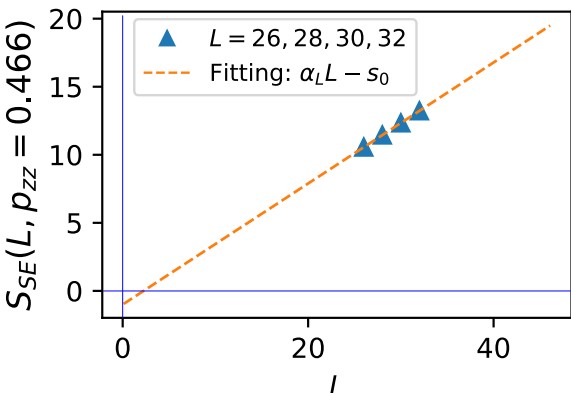

Figure 6: Fitting procedure to extract $s_0$ from different system size data of $S_{EE}$. This behavior is in the case of the XXZ model. We set the parameters $p_{zz} = 0.466$, $\Delta = 0.45$ and plot the SEE for various sets of system size $L = 26, 28, 30$ and $32$. The estimated value of $s_0$ is $s_0 = 1.0019$

As shown in this work, the SEE is a useful measure to characterize and classify mixed states with some orders. The numerical methods that we introduced in this work can be an efficient tool for discovering universality for various critical mixed states under decoherence.

(1-2)Finally, from the findings for the present the ZZ decoherence channel, we are led to the more general implication that the critical behavior of mixed states under decoherence is closely tied to the symmetry-breaking mechanism by decoherence itself. Accordingly, the value of the $g$-factor can change in a symmetry-dependent manner, suggesting a rich new type of universality classification for mixed-state criticality.

## Acknowledgements

We would like to thank the anonymous referees for their valuable comments and suggestions. This work is supported by JSPS KAKENHI: JP23K13026(Y.K.) and JP23KJ0360(T.O.).

## A    Protocol of determining $s_0$

We here show how to numerically determine the value of $s_0$ from the SEE data, $S_{SE}$. For a fixed $p_{zz}$ and $\Delta$, we calculate $S_{SE}$ for the various system sizes, and then, we plot the $L$-dependence of $S_{SE}$. In many cases, the data points exhibit behavior of a linear function of $L$. Then, we carry out the fitting procedure by assuming the fitting function, $S_{SE}(L, p_{zz}) = \alpha_L L - s_0$. By an optimization method, we can obtain $s_0$. An concrete example is shown in Fig. 6 corresponding to the data point for $p_{zz} = 0.466$ in Fig. 4 left. Here, four different system-size data exhibit a linear function behavior, then we can easily extract the estimated value of $s_0$.

## B    $e^{s_0}$ behavior for different $\Delta$ cases in XXZ model

We show the additional results for Fig. 4 left panel. We here plot the different $\Delta$ cases in Figs. 7 (a) and 7 (b). The behaviors for both cases are the same with the one shown in Fig. 4

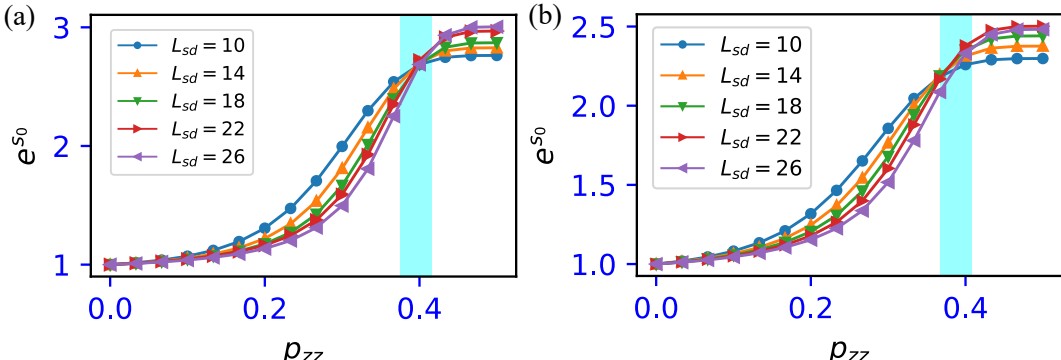

Figure 7: $p_{zz}$-dependence of the $g$-function $e^{s_0}$ for various sets of system size $\{L_{sd}, L_{sd}+2, L_{sd}+4, L_{sd}+6\}$ for critical XXZ model. (a) $\Delta = 0.15$ case. (b) $\Delta = 0.75$ case.

left panel. We also observe that data crossing for the different $L_{sd}$ data line occurs around $p_{zz} = 0.4$, that is, the crossing is independent to the value of $\Delta$. Moreover, for both $\Delta$ case shown in in Figs. 7 (a) and 7 (b), in $p_{zz} = 1/2$ limit, the value of $e^{s_0}$ tends to converge as increasing $L_{sd}$. Thus, for $L_{sd} \to \infty$, a converged value of $s_0$ exists which is independent of $\Delta$ (We assume the regime $|\Delta| < 1$).

## C  System-environment entanglement in the critical TFIM under $X + ZZ$-decoherence

As another concrete numerical example, we study the effects of the multiple decoherences to the critical state of the TFIM. This setting is considered in the previous study [39]. We consider not only $ZZ$-decoherence but also a local $X$-decoherence, the corresponding operator in the doubled Hilbert space formalism is given by

$$\hat{\mathcal{E}}_X(p_x) = \prod_{j=0}^{L-1}\left[(1-p_x)\hat{I}^*_{j,u} \otimes \hat{I}_{j,\ell} + p_x X^*_{j,u} \otimes X_{j,\ell}\right] = \prod_{j=0}^{L-1}(1-2p_x)^{1/2}e^{\tau_x X_{j,u}\otimes X_{j,\ell}}, \qquad \text{(C.1)}$$

where $\tau_x = \tanh^{-1}[p_x/(1-p_x)]$ and $0 \leq p_x \leq 1/2$.

We consider the following multiple channel

$$|\rho_D\rangle\rangle \equiv \hat{\mathcal{E}}^{ZZ}_{tot}\hat{\mathcal{E}}_X|\rho_0\rangle\rangle = C(p_{zz}, p_x, L)\prod_{j=0}^{L-1}\left[e^{\tau_{zz}\hat{h}^{zz}_{j,j+1}}e^{\tau_x\hat{h}^x_j}\right]|\rho_0\rangle\rangle, \qquad \text{(C.2)}$$

where $\hat{h}^{zz}_{j,j+1} = Z_{j,u}Z_{j+1,u}\otimes Z_{j,\ell}Z_{j+1,\ell}$, $\hat{h}^x_j = X_{j,u}\otimes X_{j,\ell}$ and $C(p_{zz}, p_x, L) \equiv (1-2p_{zz})^{L/2}(1-2p_x)^{L/2}$. Then, we expect that our target decohered state $|\rho_D\rangle\rangle$ is closely related to the ground states of the quantum Ashkin-Teller model [53], the Hamiltonian of which is given on the ladder as

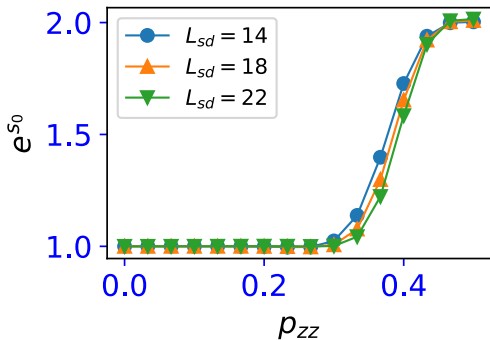

Figure 8: $p_{zz(x)}$-dependence of the $g$-function $e^{s_0}$ for various sets of system size $\{L_{sd}, L_{sd} + 2, L_{sd} + 4, L_{sd} + 6\}$ for critical TFIM model under $X + ZZ$ decoherence.

follows,

$$
\begin{aligned}
H_{\mathrm{qAT}} &= -\sum_{j=0}^{L-1}[Z_{j,u}Z_{j+1,u} + Z_{j,\ell}Z_{j+1,\ell} + \lambda_{zz}Z_{j,u}Z_{j,\ell}Z_{j+1,u}Z_{j+1,\ell}] \\
&\quad -\sum_{j=0}^{L-1}[X_{j,u} + X_{j,\ell} + \lambda_x X_{j,u}X_{j,\ell}].
\end{aligned}
\tag{C.3}
$$

The above Hamiltonian is derived from a highly-anisotropic version of 2D classical Ashkin-Teller model [54, 55] by the time-continuum-limit formalism [56], and then the Hamiltonian $H_{\mathrm{qAT}}$ has $Z_2 \times Z_2$ symmetry with generators $\prod X_{j,u}$ and $\prod X_{j,\ell}$. Furthermore, there are parameter relations such as $\lambda_{zz} \longleftrightarrow \tau_{zz}(p_{zz})$ and $\lambda_x \longleftrightarrow \tau_x(p_x)$, which are expected to qualitatively hold. The global ground state phase diagram of $H_{qAT}$ has been investigated in detail [53, 57–59]. In particular, there is a critical line in the phase diagram, which is given by $\lambda_{zz} = \lambda_x \equiv \lambda > 0$ (since $\tau_{zz(x)} > 0$) with $-1/\sqrt{2} \le \lambda \le 1$, and the criticality is described by the bosonic CFT [40]. Then, for $\lambda > 1$, a diagonal $Z_2$ symmetric phase appears (called "partially-ordered phase" [53]).

We investigate the mixed state of Eq. (C.2) under the condition of the probabilities $p_{zz} = p_x$ to realize the decoherence corresponding to $H_{\mathrm{qAT}}$ with $\lambda_{zz} = \lambda_x (= \lambda)$. Increase of $p_{zz(x)}$ corresponds to an increase of $\lambda$ in the qAT model.

Based on this setup, we numerically investigate the SEE for the state $|\rho_D\rangle\rangle$ by using the same MPS and filtering method to the main text. We also find that from the calculation of the SEE, the scaling law of Eq. (5) holds, and we extract the $g$-function $e^{s_0}$ from the data. The result as increasing $p_{zz}(= p_x)$ is shown in Fig. 8. Here, we observe that the $g$-function increases as $p_{zz}$ increases and we find that the saturation value of the $g$-function $e^{s_0}$ for $p_{zz} = 1/2$ is exactly $e^{s_0} = 2$, that is, $s_0 = \log 2$. The value of which is reminiscent of an expected value $s_0 = \log d$ with $d = 2$ proposed in [36], where the SE of the two degenerate SSB ground state of the TFIM in terms of Z-product basis and $d$ means the ground state degeneracy of the pure ground state of the TFIM.

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
