# Peer review of "System-environmental entanglement in critical spin systems under $ZZ$-decoherence and its relation to strong and weak symmetries"

_SciPost Physics Core_

## Round 1 · Referee Report · Anonymous (Referee 1) · 2025-10-6

Strengths

1) the topic is interesting 2) the paper is accessible and easy to follow

Weaknesses

1) The framework used to model decoherence is not well justified/explained 2) Numerics can be improved (with larger sizes) 3) Results are not analysed with enough detail

Report

In this paper the authors study the effect of multisite decoherence on the critical ground state of the transverse field Ising model (TFIM) and XXZ chain. They study proxies of the purity of the density matrix and they infer analogies and differences with respect to the action of a single-qubit decoherence channel.

In order to meet the Acceptance Criteria of SciPost Physics Core I feel major issues must be addressed, see below.

1) It is not clear what framework is used to model decoherence and why. From Eqs.(1,2) it seems the authors aim to model the coupling to a decoherence channel (with probability $p_{zz}$). Under certain approximations this is equivalent to the coupling to a Markovian environment for a finite time $\tau_{zz}$. However this situation seems very artificial since one is usually interested in the evolution of the system density matrix in time. Also, the discussion about the choice of the specific ZZ term is very limited. The authors briefly mentions that the reasons behind the choice has to do with strong/weak symmetries but the discussion is too short.

2) The paper miss a proper discussion about the physical meaning, interpretation and consequences of the results. The Conclusions are now a list of the results without a coherent discussion that can guide the reader. What do we learn from this work?

3) The authors use MPS methods to compute the quantities of interest. However, to the best of my knowledge, with the same methods larger sizes are easily accessible. Why do the authors do not address them? Since in the work scaling behaviours are important this would improve the overall analysis.

In summary, while the topic is promising, the manuscript in its present form does not yet meet the standards required for publication in SciPost Physics Core. Addressing the above points in a substantial revision could significantly improve the clarity and impact of the work.

Requested changes

1) Better justification of the decoherence model used. Comparison with standard Markov model leading to a Lindblad master equation. 2) Improve the conclusions with a better analysis of the results. 3) If possible extend the numerics to larger sizes. If not explain why.

Recommendation

Ask for major revision

  • validity: good
  • significance: good
  • originality: good
  • clarity: good
  • formatting: excellent
  • grammar: excellent

Author:  Yoshihito Kuno  on 2025-11-11  [id 6007]

(in reply to Report 1 on 2025-10-06)

Warnings issued while processing user-supplied markup:

  • Inconsistency: Markdown and reStructuredText syntaxes are mixed. Markdown will be used.
    Add "#coerce:reST" or "#coerce:plain" as the first line of your text to force reStructuredText or no markup.
    You may also contact the helpdesk if the formatting is incorrect and you are unable to edit your text.

To the referee 1

Thank you for reviewing our paper and suggesting useful comments. We acknowledge your useful comments, most of which have been taken into account in the revision [Please see: scipost_202509_00003v2].

In the following, we reply to the referees’ comments. * In the revised manuscript, the revised parts are highlighted with color. * In the colored revisions such as “(1-2)In this study, we investigated ~ ” in the revised manuscript, the parentheses “(1-2) ” means the revisions as the response to the number of the referees’ comments (the second comment of the first referee) and suggestions.

1-1) (a) It is not clear what framework is used to model decoherence and why. From Eqs.(1,2) it seems the authors aim to model the coupling to a decoherence channel (with probability pz). Under certain approximations this is equivalent to the coupling to a Markovian environment for a finite time τz. However this situation seems very artificial since one is usually interested in the evolution of the system density matrix in time. (b)Also, the discussion about the choice of the specific ZZ term is very limited. The authors briefly mentions that the reasons behind the choice has to do with strong/weak symmetries but the discussion is too short.

—> Thank you for these very useful comments.

(a) The ZZ-chennel can be represented by the Lindbladian form as a time evolution. Our decoherence strength parameter is related to the interval time of the dynamics.

We have added the explanation of the formulation below Eqs.(1) and (2) in the revised manuscript.

(b) The choice of the $ZZ$ operator in the channel is motivated by the symmetry aspects for the density matrix and the original pure state Hamiltonian. The system possesses global Z2 symmetry. From the theoretical perspective, we are particularly interested in decoherence operators that can induce or probe the strong-to-weak spontaneous symmetry breaking (SWSSB), which has been recently discussed in the context of unconventional mixed states. The simplest and most representative operation satisfying these requirements is the nearest-neighbor ZZ operator, which preserves the global symmetry while allowing nontrivial correlations to develop between adjacent sites.

We have added the motivation for the pick up $ZZ$ operator in the decoherence below Eqs.(1)-(2). Then, to harmonize with this change, we revised and added the sentences in the introduction. In addition, we reordered a sentence (colored) in the paragraph below Eqs.(1) and (2).

1-2) The paper miss a proper discussion about the physical meaning, interpretation and consequences of the results. The Conclusions are now a list of the results without a coherent discussion that can guide the reader. What do we learn from this work?

-->Thank you for your very variable comments. We have first revised the introduction to make it more transparent and readable for the readers. Then, we especially revised and elaborated the section of "Conclusion" by adding more discussion about what we learned from our numerical findings (See the blue colored part in Sec.7). Then, to harmonize with the change of the Conclusion, we revised and added the sentences in the introduction.

1-3) The authors use MPS methods to compute the quantities of interest. However, to the best of my knowledge, with the same methods larger sizes are easily accessible. Why do the authors do not address them? Since in the work scaling behaviours are important this would improve the overall analysis.

--> In our simulations, the main computational cost arises from the filtering MPO operation. Because we are dealing with critical systems, we performed the calculations with high numerical precision and therefore, we did not reduce the bond dimension during the filtering process. While standard DMRG simulations could in principle access larger ladder systems, our focus was on maintaining the accuracy of the non-unitary filtering procedure, which restricts the accessible system sizes smaller than that in typical MPS studies.

Nevertheless in our preliminary calculations, we have carefully examined smaller systems and confirmed that the results are already well-converged and numerically stable at the present system sizes. For this reason, we chose to perform the final simulations at these sizes with higher filtering precision.

To avoid confusion for the numerics, we have added the numerical condition for the MPS calculations in the first paragraph in Sec.5.

In summary, while the topic is promising, the manuscript in its present form does not yet meet the standards required for publication in SciPost Physics Core. Addressing the above points in a substantial revision could significantly improve the clarity and impact of the work.

--> We did our best to revise the manuscript.

---

## Round 1 · Referee Report · Anonymous (Referee 2) · 2025-10-21

Strengths

1) Interesting study of the effect of a specific kind of decoherence channel onto quantum spin chains at criticality. In principle suitable for publication in SciPost Physics Core.

2) The methods used here are adequate.

Weaknesses

1) No clear justification of the decoherence model used here.

2) A connection with possible experimental schemes is missing.

3) No specific comments on the accuracy and the actual limits of the presented numerical results.

4) Figures are too small. In some cases symbols are huge and one cannot distinguish between the various data sets.

5) The English grammar and the quality of the presentation are low.

Report

This manuscript deals with the study of decoherence in two prototypical quantum spin-chain models at criticality, namely, the transverse-field Ising chain and the spin-1/2 XXZ Heisenberg chain.
More in detail, the authors consider a two-site decoherence channel of the ZZ-type which is applied to the system, after initializing it in the ground state of the considered model. To this purpose, they employ the Choi-Jamilokovski formalism which doubles the Hilbert space, thus representing the channel as a superoperator acting on pure states in a larger space. The decohered (pure) state in such space is then numerically studied by means of MPS techniques.

They analyze the system-environmental entanglement (SSE), that is, the Rényi entropy for the density matrix of the system state, and the susceptibility of the so called Rényi-2 correlator, focusing in particular on the scaling law with L for the SSE and on the corresponding g-function.

A series of results for the Ising and the XXZ models are obtained. In particular, for the XXZ model, they find a drastic change of behavior of the g-function and of the Rényi-2 correlations in proximity of a transition caused by decoherence. Moreover the g-function under ZZ-decoherence is twice that of the same system under local Z-decoherence.

Overall, I think that the results contained here can be of some interest and may deserve publication in some form. Before proceeding with SciPost Physics Core, the authors should however consider the above concerns.

Requested changes

1) Explain why one should be interested in considering this type of two-site ZZ decoherence, rather than a more conventional single-site decoherence model. Comment this also in relation to the finding that, with two-size decoherence, the SEE is twice that with single-site decoherence.

2) Comment on the possible experimental relevance of this decoherence model, as I would be inclined to rather take a continuous time-dependent modelization of the channel, through some master equation.

3) Add specifications about the parameters that have been used in the numerical simulations with MPS. What is the typical bond-link size? Should one consider the results presented here at convergence? Is it true that L=28 is the maximum achievable system size, or can one go to larger sizes with moderate computational effort?

4) Improve the quality of the figures: in most cases, symbols are huge as compared to the size of the panels, so that it is difficult to distinguish between the various data sets.

5) The manuscript should be carefully revised in its presentation and grammar, as there are several ill defined sentences and misprints that need to be fixed. For example, at the beginning of the introduction: - "However, in research field such as quantum computers and quantum memories, effect of interactions with environment, especially decoherence, is an important research subject." - "For quantum devises such as..." - "Recently, mixed states having no pure-state counterparts in their physical properties attract lots of attention."

Recommendation

Ask for major revision

  • validity: good
  • significance: good
  • originality: ok
  • clarity: ok
  • formatting: reasonable
  • grammar: acceptable

Author:  Yoshihito Kuno  on 2025-11-11  [id 6008]

(in reply to Report 2 on 2025-10-21)

Warnings issued while processing user-supplied markup:

  • Inconsistency: Markdown and reStructuredText syntaxes are mixed. Markdown will be used.
    Add "#coerce:reST" or "#coerce:plain" as the first line of your text to force reStructuredText or no markup.
    You may also contact the helpdesk if the formatting is incorrect and you are unable to edit your text.

To the referee 2

Thank you for reviewing our paper and suggesting variable comments. We acknowledge positive comments. Please see the revised manuscript[scipost_202509_00003v2].

In the following, we reply to the referees’ comments. * In the revised manuscript, the revised parts are highlighted with color. * In the colored revisions such as “(1-2)In this study, we investigated ~ ” in the revised manuscript, the parentheses “(1-2) ” means the revisions as the response to the number of the referees’ comments (the second comment of the first referee) and suggestions.

2-1) Explain why one should be interested in considering this type of two-site ZZ decoherence, rather than a more conventional single-site decoherence model. Comment this also in relation to the finding that, with two-size decoherence, the SEE is twice that with single-site decoherence.

—> Thank you for the comment. The motivation for the choice of the $ZZ$ operator in the channel is on the symmetry aspects for the original pure state Hamiltonian. The system possesses global Z2 symmetry. From the theoretical perspective, we are particularly interested in decoherence operators that can induce some symmetry breaking (corresponding to the strong-to-weak spontaneous symmetry breaking (SWSSB)). To this end, the simplest and most representative operation is the nearest-neighbor ZZ operator, which preserves the global symmetry while allowing nontrivial correlations to develop between adjacent sites.

The above comment has been added below Eqs.(1) and (2). Then, to harmonize with this change, we have revised and added the sentences in the introduction.

Then, analogous to the spontaneous symmetry breaking in the pure state, the strong $ZZ$-decoherence for each link can lead to some cat state structure (Z2 ensemble pair) in the density matrix. Then, as a qualitative picture, we obtain a double critical mixed state leading to a possibility that the SEE is twice that with single-site decoherence.

This comment has been added in the last part of Sec.4.

2-2) Comment on the possible experimental relevance of this decoherence model, as I would be inclined to rather take a continuous time-dependent modelization of the channel, through some master equation.

—>Thank you for the comment. In fact, as the referee commented, the ZZ-chennel can be represented by the local sum of Lindbladian form as a continuous time evolution (corresponding to the master equation). For the decoherence strength parameter, we consider (p_zz) is related to the interval time of the dynamics. We have added the explanation of the formulation below Eqs.(1) and (2) in the revised manuscript. Then, the experimental implementation scheme of the local Lindbladian of the nearest-neighbor ZZ operator is feasible in a quantum circuit that was theoretically proposed in [M. Kliesch, T. Barthel, C. Gogolin, M. Kastoryano, and J. Eisert, Dissipative quantum Church-Turing theorem, Phys. Rev. Lett. 107, 120501 (2011); D. An, J.-P. Liu, and L. Lin, Linear combination of Hamiltonian simulation for nonunitary dynamics with optimal state preparation cost, Phys. Rev. Lett. 131, 150603 (2023)]

We have added the explanation of the formulation below Eqs.(1) and (2) in the revised manuscript and given a comment on the implementation in the real experimental systems and also added a new reference [[M. Kliesch, T. Barthel, C. Gogolin, M. Kastoryano, and J. Eisert, Dissipative quantum Church-Turing theorem, Phys. Rev. Lett. 107, 120501 (2011); D. An, J.-P. Liu, and L. Lin, Linear combination of Hamiltonian simulation for nonunitary dynamics with optimal state preparation cost, Phys. Rev. Lett. 131, 150603 (2023)].

2-3) Add specifications about the parameters that have been used in the numerical simulations with MPS. What is the typical bond-link size? Should one consider the results presented here at convergence? Is it true that L=28 is the maximum achievable system size, or can one go to larger sizes with moderate computational effort?

—> Thank you for the comment. We have added the numerical condition for the MPS calculations in the first paragraph in Sec.5. Please see them. This is the best achievable accuracy and system size within our available computational resources.

In our simulations, the main computational cost arises from the filtering MPO operation. Because we are dealing with critical systems, we performed the calculations with high numerical precision and therefore, we did not reduce the bond dimension during the filtering process. While standard DMRG simulations could in principle access larger ladder systems, our focus was on maintaining the accuracy of the non-unitary filtering procedure, which restricts the accessible system sizes smaller than that in typical MPS studies.

2-4) Improve the quality of the figures: in most cases, symbols are huge as compared to the size of the panels, so that it is difficult to distinguish between the various data sets.

—> Thank you for the comment. We replot figures 1 and 3 to eliminate the difficulty. We also enlarged some figures.

2-5) The manuscript should be carefully revised in its presentation and grammar, as there are several ill defined sentences and misprints that need to be fixed. For example, at the beginning of the introduction: - "However, in research field such as quantum computers and quantum memories, effect of interactions with environment, especially decoherence, is an important research subject." - "For quantum devises such as..." - "Recently, mixed states having no pure-state counterparts in their physical properties attract lots of attention."

—>Thank you for the comments We have made detailed corrections to the wording and grammar throughout the entire manuscript. We made careful revisions to the introduction and conclusion, focusing on improving English usage and grammatical accuracy.

---

## Editorial Decision

in_refereeing